# Numerical Prediction of Erosion Based on the Solid-Liquid Two-Phase Flow in a Double-Suction Centrifugal Pump

Xijie Song [1] , Rao Yao [1], Yubin Shen [2], Huili Bi [1], Yu Zhang [2], Lipu Du [2] and Zhengwei Wang [1,*]

1   Department Energy & Power Engineering, Tsinghua University, Beijing 100084, China;
    songxijie@mail.tsinghua.edu.cn (X.S.); yaorao19@mails.tsinghua.edu.cn (R.Y.);
    bihuili2014@mail.tsinghua.edu.cn (H.B.)
2   Water Conservancy Project Construction Center of Ningxia Hui Autonomous Region,
    Yinchuan 750000, China; 18795009750@139.com (Y.S.); 18252739209@139.com (Y.Z.);
    15253410359@139.com (L.D.)
*   Correspondence: wzw@mail.tsinghua.edu.cn

**Abstract:** Due to the high sediment content in the Yellow River, the pump units in the pumping stations along the line are often eroded by sediment, causing the reduction of pump efficiency and structural damage. The purpose of this paper is to study the influence of particle diameter on the particle track, erosion distribution and erosion rate in a double-suction centrifugal pump in a pumping station of the Yellow River with a Lagrangian particle-tracking approach and a Tabakoff erosion model. The results show that the surface erosion of the impeller on both sides in the double-suction centrifugal pump has an asymmetric distribution, and the erosion rate on both sides is different. The particle diameter affects the moving trajectory of particles and has a significant effect on the erosion morphology and position in the impeller. With the increase of particle diameter, the velocity of the particles moving towards the pressure side of the blade inlet increases, resulting in punctate impact erosion. When the particle diameter decreases, sliding abrasion gradually forms on the pressure side of the blade outlet. The change rule of the solid particle volume fraction on the impeller wall is consistent with that of erosion distribution on the impeller wall. The larger the solid volume fraction is, the higher the wall erosion rate is.

**Keywords:** centrifugal pump; impeller; erosion; particle track; particle diameter

## 1. Introduction

The Yellow River has the highest sediment concentration in the world. Many large pumping stations have been built along it for irrigation and urban water supply [1]. Double suction centrifugal pumps have the characteristics of large flow, high head and good cavitation performance [2]. Therefore, double suction centrifugal pumps are used in more than 70% of the pumping stations along the Yellow River [3]. Due to their high content, the sediment particles have serious erosion and wear on the flow passage of the pumps, which seriously affects the normal operation, reduces the working efficiency and has a great impact on the normal production [4,5].

At present, a large amount of literature has studied and analyzed the solid-liquid two-phase flow in centrifugal pumps, with especially profound research upon the erosion and damage of the parts of the centrifugal pump flow passage [6]. Xu [7] used a high-speed camera to observe the motion regulation of solid particles in a centrifugal pump impeller, and obtained that the particle diameter and its density in the fluid, impeller speed and blade angle have obvious influence on the particle motion. Liu [8] found that the properties of discrete particles and impeller speed have important influence on the trajectory of solid particles and the process of wall collision by tracking the particle trajectory in a solid-liquid flow field and numerical simulation of solid-liquid two-phase flow. Qian [9] found that

the degree of the blade wear is related to velocity and impact angle through numerical simulation of different blade inlet edge shapes.

For the numerical simulation of solid-liquid two-phase flow, the Mixture model was used for simulation [10]. In recent years, there much literature has been produced on the simulation of solid-liquid two-phase flow using the particle track model, which has solutions for the most complex problems, especially the discrete solid particle flow [11]. The particle track model takes particles as discrete phase and liquid as continuous phase, and calculates particle motion in a Lagrange coordinate system and continuous phase motion in an Eulerian coordinate system [12]. Then, a large number of particles are counted to obtain the macro trajectory of particle motion. The velocity of particles on any trajectory can be obtained by the Lagrange method. Xu [13] used this method to simulate the movement of solid particles in the centrifugal pump, and the predicted results are close to the experimental results.

Meanwhile, in order to ensure the reliability of erosion rate prediction, Finnie [14], Tabakoff [15], Engin [16] and Khurana et al [17] have proposed different erosion rate prediction models. These empirical models are basically established on the basis of erosion tests, and the actual variables (impact velocity, impact angle and impact times) determine the value of these variables. There are two default erosion prediction models in CFX (Computational Fluid Dynamics X), the Finnie [18] erosion model and Tabakoff erosion model [19]. There is some literature that shows the calculation results adopted the Tabakoff erosion model are more accurate by comparing with the experimental results [20,21].

Due to the complexity of the flow in the rotating machinery and the high sediment content of the pumping station along the Yellow River, there are relatively few studies on the erosion of the prototype double-suction centrifugal pump. In this paper, the purpose is to study the influence of the particle diameter on the particle track, erosion distribution and erosion rate in a double-suction centrifugal pump in a pumping station of the Yellow River with a Lagrangian particle-tracking approach and a Tabakoff erosion model, which is helpful to solve the problem of abrasion damage in engineering.

## 2. Numerical Calculation

### 2.1. Research Object

The research object of this paper is a double-suction centrifugal pump. The main parameters of the whole calculation model are: rated flow 3.083 m$^3$/s, rated head $H = 50$ m, rated efficiency $\eta_r = 86\%$, rotating speed $n = 490$ r/min, blade number $Z_b = 16$, impeller diameter $D = 1275$ mm and the clearance between volute and impeller is 4 mm. Figure 1 shows the double-suction centrifugal pump. Figure 2 shows the name of impeller components.

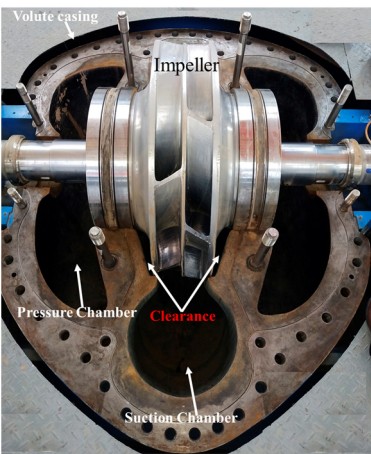

**Figure 1.** Double-suction centrifugal pump.

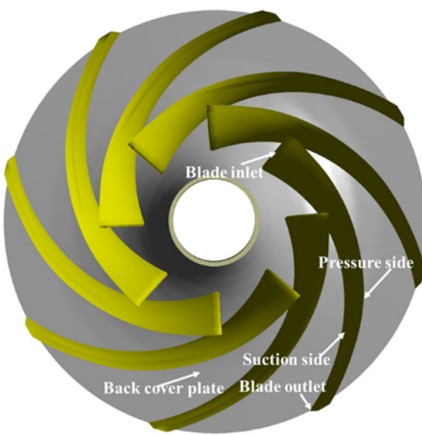

**Figure 2.** Diagram of the impeller.

The Workbench is used to mesh the calculation model, and the unstructured grid and structured grid are selected for hybrid generation, as shown in Figure 3. The flow velocity near the wall directly affects the surface erosion. To ensure the fine calculation of surface erosion and the y+ value within 100, the boundary layer is set on the wall, according to the inlet Reynolds number $Re = \rho U_\infty l / \mu = 2.35 \times 10^6$. There are 20 layers in the boundary layer. The height of the first boundary layer on the impeller wall is $2.13 \times 10^{-2}$ mm. The grid size of the whole unit is divided from coarse to fine, and the pump head is used to verify the grid independence, as shown in Figure 4. When the number of grids is 4.21 million, the change of pump head is within 1.8%, so the number of grids is 4.21 million.

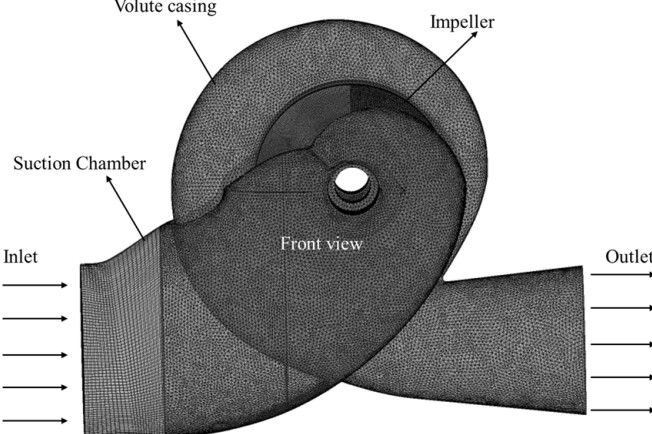

**Figure 3.** Calculation model grid.

### 2.2. Mathematical Model for Solid-Liquid Two-Phase Flow

In sediment-laden flow engineering, slurry flow can be treated as diluted flow, and the volume fraction of particles in these slurry flows is not very high [22]. In this study, the two-way coupling method is used for the model, ignoring the collision between particles. The momentum exchange between discrete and continuous phases makes the particles move [23]. The Euler method is used to calculate the continuous phase liquid, and the Lagrange method is used to calculate the position and velocity of discrete phase particles in the flow field.

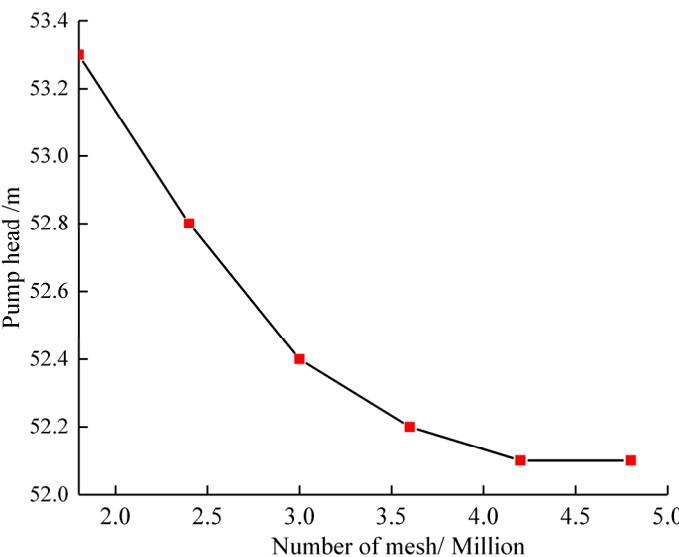

**Figure 4.** Grid independence analysis.

### 2.2.1. Mathematical Model for Liquid Phase

$$\frac{\partial \rho_l}{\partial t} + \frac{\partial (\rho_l u_j)}{\partial x_j} = 0 \tag{1}$$

$$\frac{\partial (\rho_l u_j)}{\partial t} + \frac{\partial (\rho_l u_i u_j)}{\partial x_j} = -\frac{\partial p}{\partial x_j} + \frac{\partial}{\partial x_j}\left[(\mu_l + \mu_t)\left(\frac{\partial u_i}{\partial x_j} + \frac{\partial u_j}{\partial x_i}\right)\right] + Su_{ip} + \rho g_i \tag{2}$$

where the $u_i$ and $u_j$ is the liquid velocity component at *i* direction and *j* direction, $\rho$ is the liquid density, $\mu_t$ is the turbulent viscosity, $\mu_l$ is the fluid viscosity, and *p* is the mean pressure. The phase interactions are considered by the additional source terms $Su_{ip}$, which are incorporated in the Navier-Stokes equations and calculated implicitly.

### 2.2.2. Basic Equation of Particle Motion

In the Lagrangian framework, when the particle moves in the liquid, the force on the particle comes from the velocity difference between the particle and the fluid [24]. The main forces on particles are gravity, resistance, virtual mass force, pressure gradient force, Basset force, Saffman force, Magnus force and so on. The basic equation of particle motion is as follows:

$$m_p \frac{du_p}{dt} = F_D + F_B + F_G + F_V + F_P + F_X \tag{3}$$

where *t* is time, $m_p$ is particle mass, $u_p$ is particle velocity, $F_D$ is drag force, $F_B$ is Basset force, $F_G$ is gravity, $F_V$ is virtual mass force, $F_P$ is pressure gradient force and $F_X$ is the sum of other external forces considered.

In this paper, the particle concentration in the flow field is small, the fluid velocity of the continuous phase in the pump is large, and there is a large density difference between the continuous phase and the discrete phase. Therefore, the virtual mass force, pressure gradient force, Basset force, Saffman force and Magnus force on the solid particles can be ignored. The basic equation of particle motion can be expressed as:

$$\frac{dx_{pi}}{dt} = u_{pi} \tag{4}$$

$$\frac{du_{pi}}{dt} = \frac{3C_{D}\rho f}{4\rho_p D_p}|u_s|u_s \tag{5}$$

where, $u_s$ is the slip velocity between particles and liquid, $C_D$ is the drag coefficient related to the Reynolds number, $\rho_f$ is the liquid density, $\rho_p$ is the particle density, $D_p$ is the particle diameter and $x_{pi}$ is the spatial coordinate position of particles.

It can be seen from the basic equation of particle motion that when a particle moves in the liquid, its trajectory is related to the particle diameter and density.

### 2.3. Erosion Model

ANSYS CFX is used to calculate the flow field. The particle track model is used to calculate the flow field. The coupling mode of the calculation model is one-way coupling. The Tabakoff erosion model is used to predict erosion. According to the particle track model, each group of particles moves along its own independent trajectory from the initial position, the particles are independent of each other and there is relative velocity slip between particles and fluid [25]. The turbulent diffusion, viscosity and heat transfer between particles are ignored.

The Tabakoff erosion model used to predict erosion is an empirical and semi-empirical erosion model based on the study of the effects of different particle velocities and collision angles on target erosion [26,27]. The model is calculated based on the angle and velocity at which particles collide with the impeller (that is particle trajectories). The formula is as follows:

$$E = f(\gamma)\left(\frac{V_p}{V_1}\right)^2 \cos^2\gamma \left[1 - \left(1 - \frac{V_p}{V_3}sin\gamma\right)^2\right] + \left(\frac{V_p}{V_2}sin\gamma\right)^4 \qquad (6)$$

$$f(\gamma) = \left[1 + k_1 k_{12} \sin\left(\gamma\frac{\pi/2}{\gamma_0}\right)\right]^2 \qquad (7)$$

$$k_1 = \begin{cases} 1 & \gamma \leq 2\gamma_0 \\ 0 & \gamma > 2\gamma_0 \end{cases} \qquad (8)$$

here $E$ is the dimensionless mass (mass of eroded wall material divided by the mass of particle). $V_p$ is the particle impact velocity. $V_1$, $V_2$ and $V_3$ are the parameters of particle impact velocity. $\gamma$ is the impact angle in radians between the approaching particle track and the wall, $\gamma_0$ being the angle of maximum erosion. $k_1$, $k_{12}$ and $\gamma_0$ are model constants and depend on the particle/wall material combination.

Equation (4) can be divided into two parts: the first is the small angle cutting damage caused by particles, which is the damage mechanism of particles to expansible materials. The second is the erosion damage to the target caused by the particles along the direction of the normal velocity, which is proportional to the fourth power of the velocity, which is the damage mechanism of the particle to the brittle material.

Because the erosion model can comprehensively consider the joint effects of ductile materials and brittle materials, it can more comprehensively predict the erosion characteristics. However, due to the large number of empirical coefficients and strong pertinence for the materials, the erosion model is mainly suitable for steel, aluminum and other materials. Since the number of grids reaches 4.21 million and the calculation medium is solid-liquid two-phase flow, parallel calculation with 20 calculation nodes takes about 26 h for a single working condition.

### 2.4. Solution Calculation Method

The calculated boundary conditions are shown in Figure 3. The total pressure inlet was set as the inlet boundary condition, the mass flow outlet was prescribed as the outlet boundary condition with the designed flow 3.083 m$^3$/s and the inlet particle volume fraction is assumed to be evenly distributed. For the wall in contact with liquid in the flow passage part, the non-slip wall condition is adopted for the fluid phase, the free slip wall condition is adopted for the solid particle phase and the standard wall function is adopted near the wall. The turbulence model adopts the SST k-ω model (shear-stress transport k-ω

turbulence model). The particle track model is used for solid phase, and the Tabakoff and Grant erosion model is used for the erosion model. The high precision difference scheme and RMS (Residual Mean Square) residual scheme are used to solve the problem, and the accuracy is set to $10^{-5}$.

In this paper, the movement of solid particles with different particle diameters in liquid is analyzed. In a CFX setting, the discrete term of particles is set as dilute phase, and the collision between particles is not considered. The collision between the particle and the solid wall is completely elastic without considering the energy loss.

## 3. Results and Discussions

### 3.1. Calculation Scheme and Reliability Verification

The variation of particle diameter of the Yellow River in different periods is quite broad. This paper mainly studies the influence of particle diameter on particle movement, erosion distribution, erosion rate and solid volume fraction distribution.

Shen et al [19] studied the particle size distribution in a pumping station forebay of the Jingtai Yellow River Irrigation Project (JYRIP) using SEM micrographs. All sediment examples have sizes of less than 250 μm, 97% of sand particles had size less than 100 μm and 30% of sand particles had sizes that were less than 4 μm. The mean diameter of the sand particles entering the pump is 25 μm. Both the pumping station in this paper and JYRIP are located in the middle and upper reaches of the Yellow River, and the sediment condition belongs to the same category. So the calculation is carried out under the conditions of 0.01 mm, 0.025 mm, 0.05 mm and 0.2 mm of the average annual sediment concentration of the Yellow River at 15 kg/m³. The specific sediment conditions are shown in Table 1.

**Table 1.** Sediment condition.

| Parameter | Scheme Number | | | |
|---|---|---|---|---|
| | **1** | **2** | **3** | **4** |
| Particle diameter (mm) | 0.01 | 0.025 | 0.05 | 0.2 |
| mass concentration (15 kg/m³) | 15 | 15 | 15 | 15 |

Figure 5 shows the change of energy characteristics of the pump curve obtained by numerical simulation and experimental testing. The experimental data of energy characteristics are provided by Andritz company. The variation trend of energy characteristics obtained by numerical simulation is consistent with that obtained by experiment, and the error is within 0.3%.

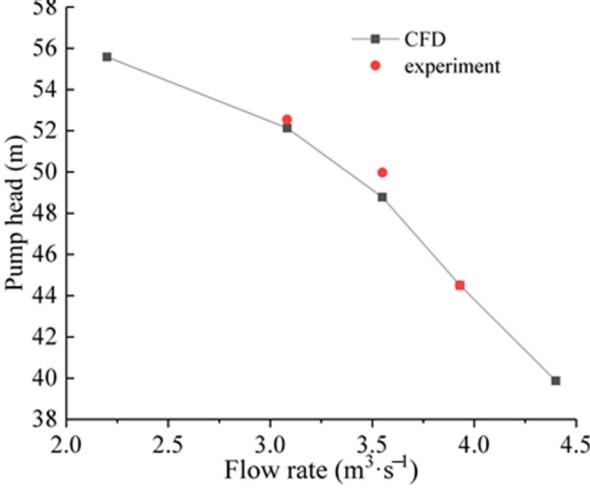

**Figure 5.** Verification of energy performance of numerical simulation pump.

Figure 6 is a physical picture of the abrasion of the pump unit of the Yellow River Pumping Station. Figure 7 shows the distribution of erosion rate when the particle inlet mass concentration is 15 kg/m$^3$ and the particle diameter is 0.025 mm.

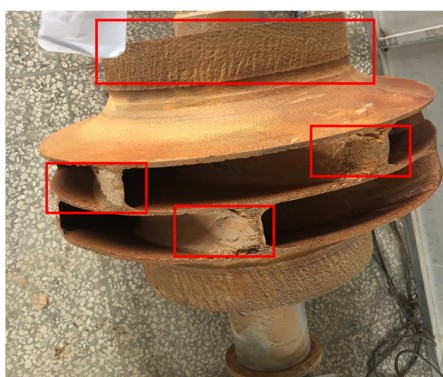

**Figure 6.** Physical picture of erosion.

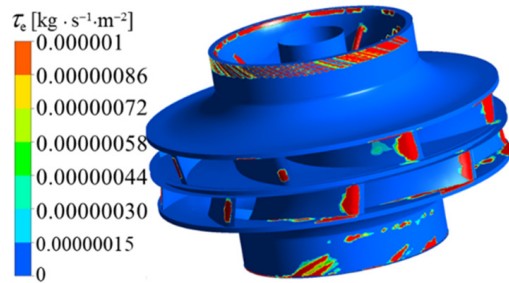

**Figure 7.** Erosion result of numerical simulation.

In fact, the erosion in the pump is due to long-term damage caused by different factors, and numerical simulation is an ideal method in the calculation prediction of erosion and the actual erosion, but it is inevitable there will be some errors. On the whole, the erosion position and erosion pattern of the front cover wall and the blade outlet pressure side predicted by the numerical simulation are consistent with the actual erosion characteristics of the water pump impeller on the project site. The calculation results of energy characteristics and erosion characteristics show that the numerical simulation method is reliable. In addition, this study is based on existing engineering problems, and the relevant research results will be applied in engineering.

Due to the erosion of the blade, the operation efficiency of the pump will decrease greatly. This paper mainly analyzes the erosion characteristics of the blade surface and the change of particle trajectory in the impeller.

*3.2. Regular Patterns Movement of Particles with Different Diameters in Double Suction Centrifugal Pump*

The particle trajectories of different particle diameters, with the mass concentration of particles being 15 kg/m$^3$, are shown in Figure 8. On the whole, the particle tracks on both sides of the double suction centrifugal pump have an asymmetric distribution. With the increase of particle diameter, the slip velocity of particles in the impeller also increases, and its movement trend to the blade's pressure side is more obvious. The collision with the blade gradually approaches to the inlet edge, and then slides along the junction of the blade and the back cover plate to the outlet edge of the blade.

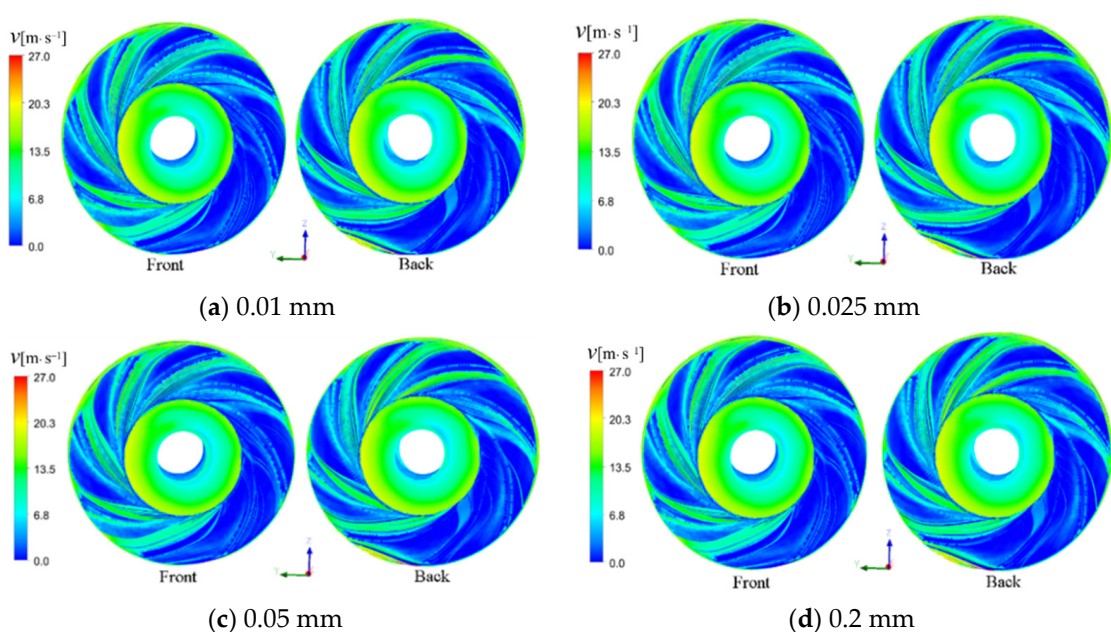

**Figure 8.** Particles tracks with different diameters.

The volume fraction of solid particles on the impeller wall determines the erosion degree of particles on the impeller wall. The larger the volume fraction, the more times the solid particles wear the wall. The slip velocity of solid particle phase on the wall directly determines the erosion ability of single particle on the wall. Figures 9 and 10 show the volume fraction distribution and particle distribution of different particle diameters in the impeller back cover and blade passage, respectively. With the increase of particle diameter, the volume fraction of solids at the inlet of blade increases gradually, and the volume fraction of solids at the pressure side at the outlet of blade also increases gradually, which is consistent with the movement law of particle trajectory. When the particle diameter is larger, the volume fraction on the wall tends to 1. Considering the slip velocity and volume fraction distribution of solid particles, it is concluded that with an increase of particle diameter, the erosion degree of the blade's pressure side increases correspondingly.

On the whole, the slip velocity of the particles at the impeller outlet is obviously larger, which will cause rapid erosion on the wall of the volute. The larger the particle diameter is, the more serious the erosion is; this is because the larger the particle diameter is, the greater the inertial force is. When the diameter of the solid phase is small, its distribution is relatively uniform. With the increase of particle diameter, the volume fraction of solid particles at the inlet of impeller increases gradually, and the volume fraction of solid particles at the pressure side of the impeller outlet also increases gradually, which is consistent with the regular patterns of particle trajectories. At the inlet of the pressure side, the volume fraction of solid particles is the largest. Considering the slip velocity and volume fraction distribution of solid particle phase, it is concluded that the volume fraction is obviously affected by particle diameter. With the increase of particle diameter, the erosion degree of the pressure side increases correspondingly.

### 3.3. Particle Tracks with Different Diameters in a Single Passage

The particle tracks with different diameters in a single passage of impeller are observed, as shown in Figure 11. The particles in the single blade channel are mainly concentrated in the inlet and outlet sides of the blade. The particles enter the blade channel from the impeller inlet and carry a large amount of sediment to impact the suction surface wall of the adjacent blade outlet from the impeller inlet in the form of the bias flow, causing sliding friction and erosion on the suction surface wall.

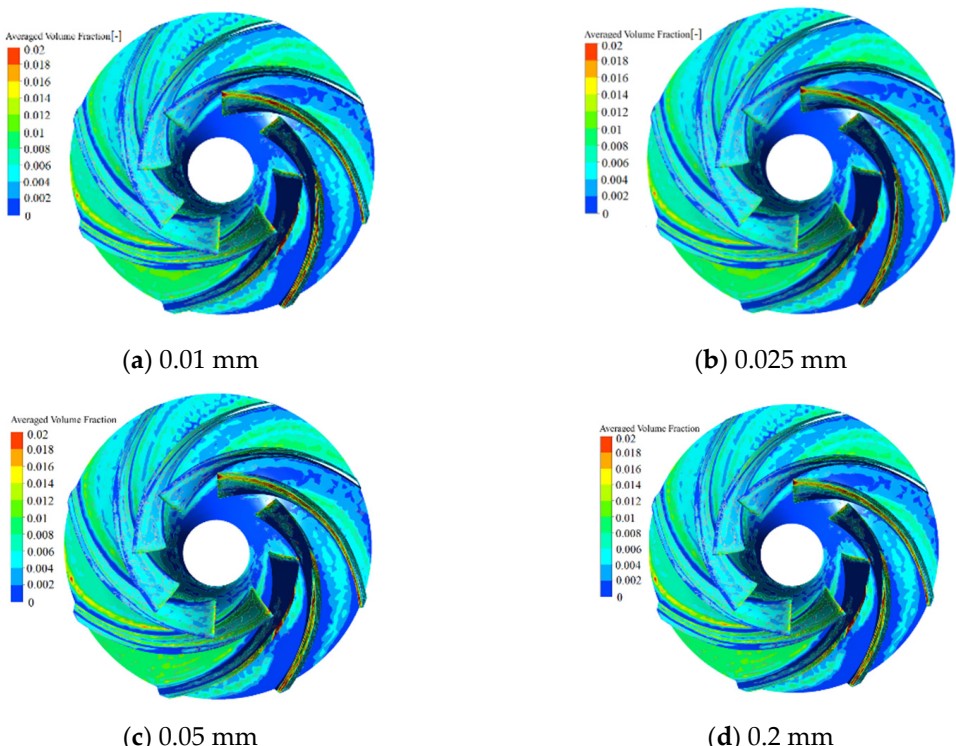

(**a**) 0.01 mm　　　　　　　　　　　　　　　(**b**) 0.025 mm

(**c**) 0.05 mm　　　　　　　　　　　　　　　(**d**) 0.2 mm

**Figure 9.** Volume fraction distribution of solid particle phase in impeller (front view).

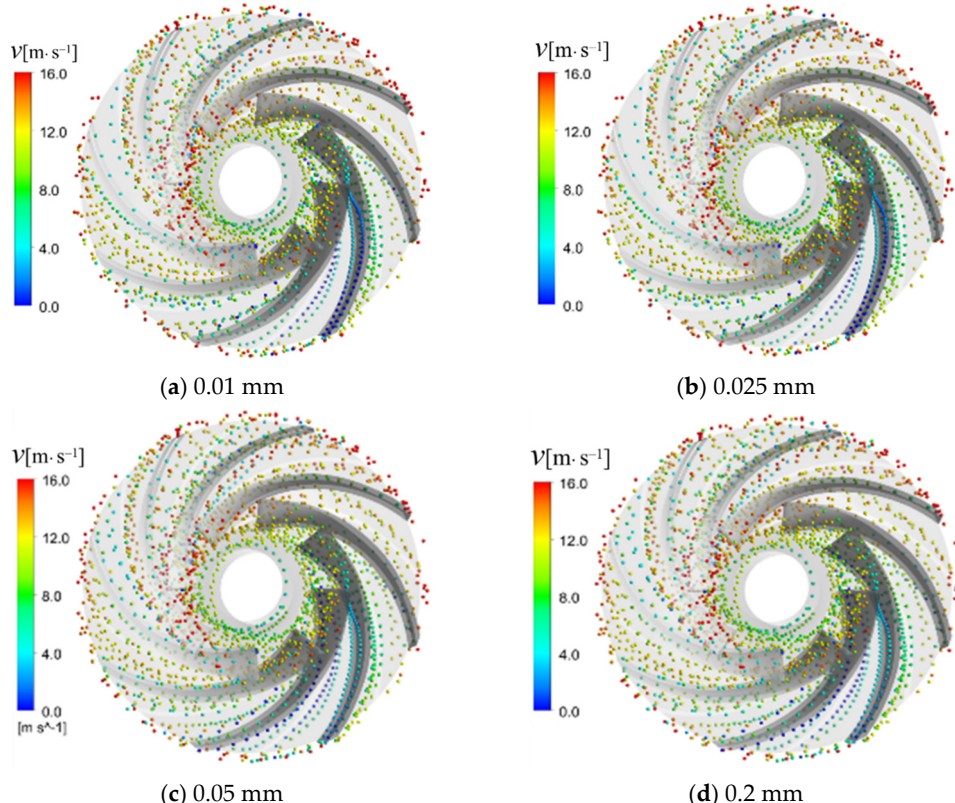

(**a**) 0.01 mm　　　　　　　　　　　　　　　(**b**) 0.025 mm

(**c**) 0.05 mm　　　　　　　　　　　　　　　(**d**) 0.2 mm

**Figure 10.** Distribution of particles with different diameters in impeller (front view).

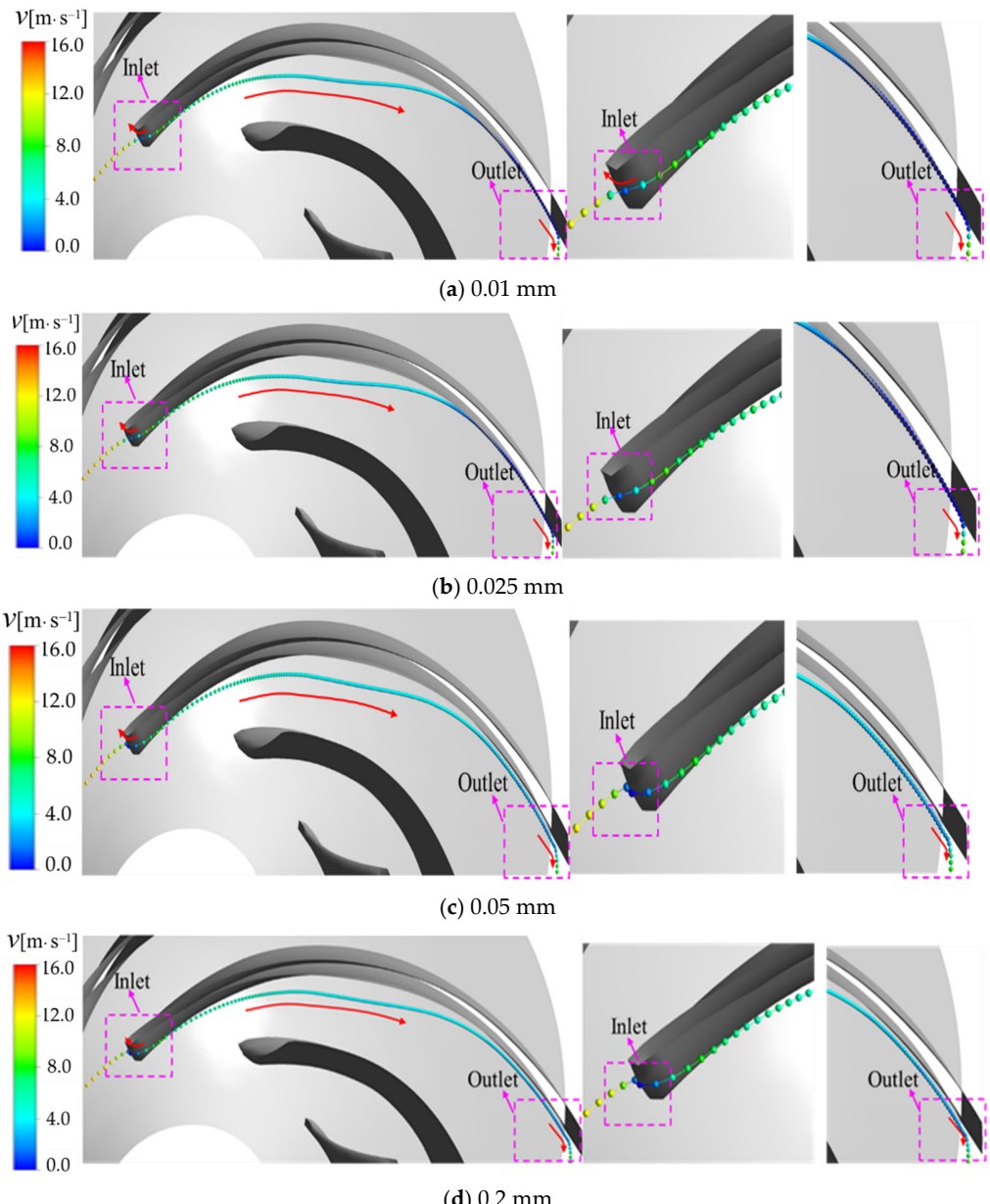

**Figure 11.** Particle tracks with different diameters in a single passage (front view).

The particles flow along the suction side to the outlet. After sliding along the suction side near the inlet for a certain distance, particles gradually separate from the blade wall. They slide along the wall again when the particles reach the suction side near the impeller outlet. The particles have no contact with the center of the blade's suction side.

The particles with small particle diameters follow the flow well, which flows along with the water towards the suction surface, and the contact range with the suction surface of the blade outlet is large. With the increase of particle diameter, the influence of flow field on particle trajectory is weakened, but the influence of inertia and centrifugal force is increased.

At the blade outlet, the larger the particle diameter, the easier the particles deviate from the blade wall. At the blade inlet, the larger the particle diameter is, the larger the deflection trend of the particle to the pressure side of the blade is and it is easier for the particle to collide with the blade inlet. Small particles easily collide with the pressure side at the blade outlet, but the probability of secondary collision at the same position is small.

### 3.4. Erosion Analysis of Impeller with Different Particle Diameter

Figures 12 and 13 show the distribution of erosion rate on the whole impeller and single blade with different particle diameters under the particle concentration of 15 kg/m³. On the whole, the erosion distribution and rate on both sides of the impeller is asymmetric. Affected by the uneven distribution of inlet flow, the erosion rate on both sides is different, and the erosion on the side with high sediment content is serious.

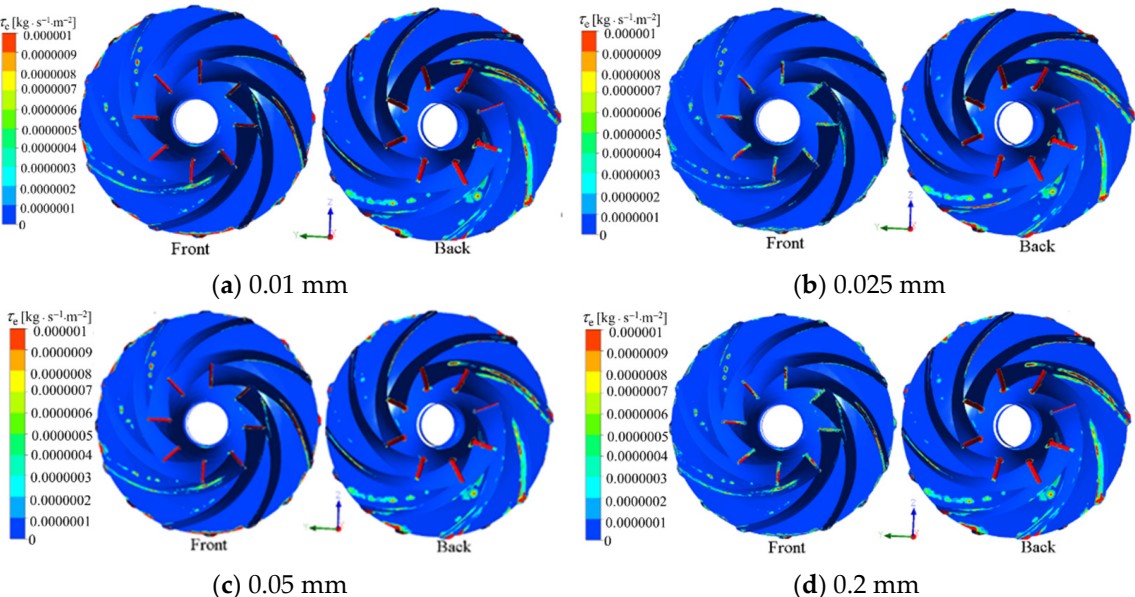

**Figure 12.** Distribution of erosion rate on impeller with different particle diameters.

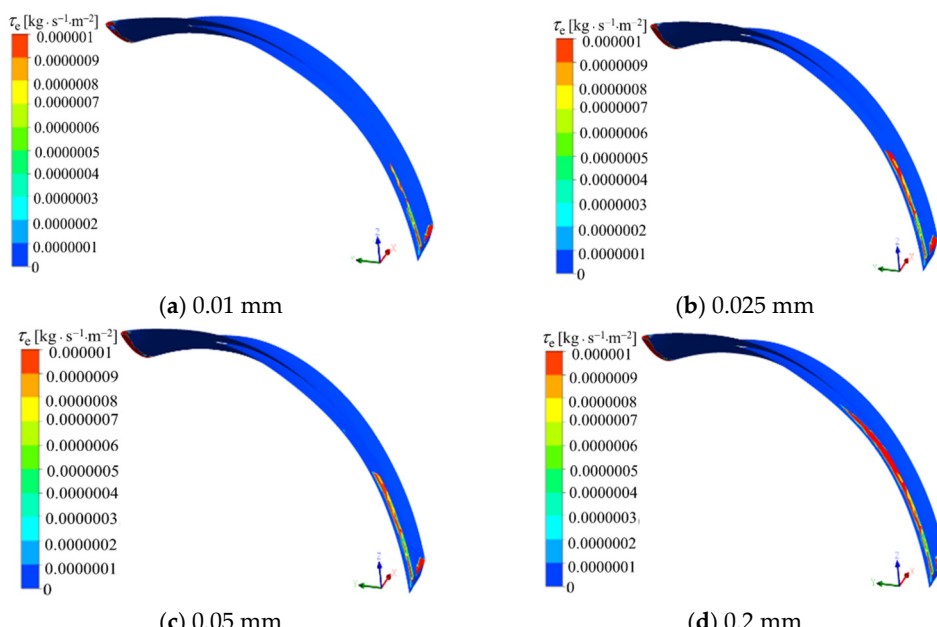

**Figure 13.** Distribution of erosion rate on the single blade with different particle diameters.

The erosion in the impeller is mainly concentrated in the back cover plate, the blade inlet and the blade outlet. When the particle diameter is small, the erosion mainly occurs near the blade outlet due to the slow movement of particles to the blade, and the erosion shape is continuous strips, which is caused by the sliding friction between the small particles and the blade wall.

With the increase of particle diameter, the erosion position of particles in the blade gradually approaches to the pressure side at the blade inlet, and the erosion at the blade inlet extends to the pressure side. This is because, with the increase of particle diameter, the moving velocity of particles in the impeller increases, the slip velocity of particles moving to the pressure side increases and the collision between particles and blades gradually approaches the blade inlet. This is consistent with the phenomenon described by particle trajectory. In addition, the erosion shape of the larger particles near the inlet is in the form of dots, which is caused by the collision of the larger impact velocity carried by the larger particles and the blade inlet, indicating that the erosion shape is related to the particle diameter.

In this paper, a quantitative analysis is made on the erosion amount at the blade inlet and blade of a single blade. It is assumed that after one year of continuous operation of the pump, the erosion amount change of a single blade at the inlet and tail under different particle diameters is analyzed, as shown in Figure 14. Under the conditions of particle diameters of 0.01 mm, 0.025 mm, 0.05 mm and 0.2 mm, the erosion amount at the inlet edge of blade is 2.32 kg, 2.89 kg, 3.12 kg and 4.21 kg, respectively, and the erosion amount at the outlet edge of blade is 4.76 kg, 5.36 kg, 5.89 kg and 6.82 kg, respectively.

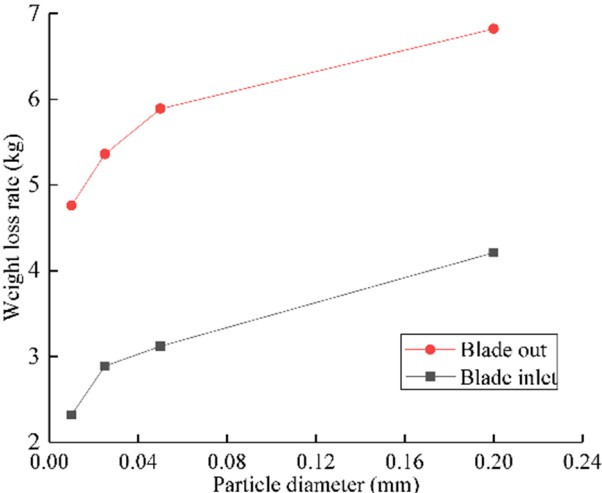

**Figure 14.** The erosion amount change under different particle diameters.

On the whole, the amount of erosion on the blade outlet is greater than that on the blade inlet. This is mainly due to the large slip velocity of the particles on the outlet side and the serious erosion of the particles on the wall surface. The changes in the amount of erosion at the two positions indicate that with the increase of the particle diameter, the total amount of erosion at the inlet and outlet of the blade increases. This is consistent with the phenomenon described by the erosion distribution; that is, the larger the particle diameter, the larger the erosion area and the greater the amount of erosion.

We also compare the operating efficiency changes under three typical conditions of 0.75 $Q_d$, $Q_d$ and 1.2 $Q_d$ under different particle diameters to further quantitatively analyze the influence of particle diameters on pump performance, as shown in Figure 15. On the whole, the operation efficiency of the pump with sediment-laden water is lower than that with pure water. When the particle size is 0.01 mm and 0.025 mm, the efficiencies of the pumps are close to each other. When the particle size is 0.2 mm, the efficiency of the pump decreases the most, indicating that the larger the particle diameter is, the greater the impact on the operation of the pump is, which is consistent with the variation rule of the surface erosion of the pump under different particle diameters.

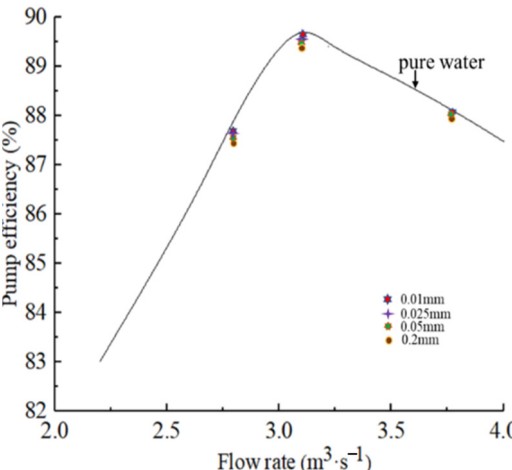

**Figure 15.** Pump efficiency under different particle diameters.

## 4. Conclusions

Sediment erosion is a key problem affecting the performance and service life of double-suction centrifugal pump operation. This paper analyzes the influence of particle distribution and particle trajectory change on blade surface erosion with different particle diameters. The conclusions are as follows:

(1) The erosion distribution of the impeller surface on both sides of a double-suction centrifugal pump is asymmetric. Affected by the uneven distribution of inlet flow, the erosion rate on both sides is different, and the erosion on the side with high sediment content is serious. The erosion in the impeller of a double suction centrifugal pump is mainly distributed in the back cover plate, blade pressure side and suction face tail.

(2) The particle diameter affects the moving trajectory of particles in the pump. The particles with a small particle diameter follow the flow well; the particles flow along the suction surface with the flow and the contact range between the suction surface and the wall at the blade outlet is large. With the increase of particle diameter, the influence of flow field on particle trajectory is weakened, and the influence of inertia and centrifugal force is enhanced. The larger the particle diameter at the blade outlet, the easier the particles deviate from the blade wall. At the blade inlet, the larger the particle diameter is, the larger the deflection trend of the particle to the pressure side of the blade is, and it is easier for the particle to impact with the blade inlet. Small particles easily collide with the pressure side at the blade outlet, but the probability of secondary collision at the same position is small. With the increase of the particle diameter, the slip velocity of particles on the wall of the flow passage increases.

**Author Contributions:** Data curation, X.S.; software, R.Y.; validation, H.B.; formal analysis, Y.Z.; investigation, Y.S.; resources, L.D.; writing-original draft preparation, X.S.; writing-review and editing, Z.W. All authors have read and agreed to the published version of the manuscript.

**Funding:** This work was supported by Water conservancy science and technology projects in Ningxia Hui Autonomous Region [DSQZX-KY-01, DSQZX-KY-02], Joint open fund of Tsinghua University Ningxia Yinchuan water network digital water control joint research institute (sklhse-2021-Iow10). National Natural Science Foundation of China (51876099).

**Institutional Review Board Statement:** Not applicable.

**Informed Consent Statement:** Not applicable.

**Data Availability Statement:** Not applicable.

**Conflicts of Interest:** The authors declare no conflict of interest.

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
