# Peer review of "Numerical Prediction of Erosion Based on the Solid-Liquid Two-Phase Flow in a Double-Suction Centrifugal Pump"

_jmse, doi:10.3390/jmse9080836_

Round 1

Reviewer 1 Report

Broad comments. The authors have made a concise overview of the topic and a brief reference to existing literature. They have indicated the main task of the paper among its motivation. Finally, they have pointed out the key message and the potential benefits of their work. As a general drawback, I could say that there is no validation effort for the results.

Specific comments. In general, the text is very well structured and has clearly defined topics. The abstract is a very good guide for what follows. More or less all fundamental theory details that are needed are discussed and concluding remarks are sufficient. Some comments for improvement:

  1. Authors could describe better the difficulties of the study based on the dual state of Solid-Liquid Two-Phase Flow. Parametric studies on similar phenomena are crucial and this is part of this effort’s innovation.
  2. Authors could explain the decision/assumptions regarding the forces included in Equation 1. Due to the highly non-linear nature of these forces/moments solution of the relevant equation could be very demanding.
  3. Authors could specify relevant computational cost in section 2.3.
  4. In general, all acronyms should be explained the first time used, even if their use is trivial (e.g. SEM, RMS, SST, CFX).
  5. Authors provide initial verification of their methodology based on energy conservation. Verification results either by comparison to other numerical results (different authors, different methods, etc) or by comparisons with experimental results would be more than welcome. This will elevate the impact of the study.

Author Response

Dear  Reviewers:

Thank you for your letter and for the reviewers’ comments concerning our manuscript entitled “Numerical prediction of erosion based on the solid-liquid two-phase flow in a double-suction centrifugal pump” (ID: jmse-1265719). Those comments are all valuable and very helpful for revising and improving our paper, as well as the important guiding significance to our researches. We have studied comments carefully and have made correction which we hope meet with approval. Revised portion are marked in the paper. The main corrections in the paper and the responds to the reviewer’s comments are as flowing:

Response to comments 1 and 2 :

The reviewer’s comments are very correct and objective, and the study on“the difficulties of the study based on the dual state of Solid-Liquid Two-Phase Flow”and “the decision/assumptions regarding the forces in the equation of particle motion”is carried out in another paper being submitted.

  1. Response to comments 3 :

The relevant computational cost has been added in section 2.3.

  1. Response to comments 4 :

All acronyms have been explained

  1. Response to comments 5 :

Fig. 7 and Fig. 8 are the physical and numerical erosion diagrams of pump erosion in engineering, respectively. The calculated erosion distribution is consistent with the actual erosion position and shape of pump. Relevant quantitative test data will be tested in the project.

Reviewer 2 Report

Writing. The authors need to read-proof the paper.

Mesh Quality check. It is necessary to add a quality check process and its results (e.g., y+).  Also, how the influences of rotation and curvature are taken into account; and the coupling between impeller and volute is implemented?

References. The authors need to up-date the reference section, by including more relevant references.

Mathematical model. In section 2, a mathematical model for the solid-liquid two-phase flow could be added.

Results.

- Figure 6. The experimental data of energy characteristics are provided by Andritz company. Need to provide the reference.

- This study investigated the influence of the particle diameter on the particle track, erosion distribution and erosion rate in a double-suction centrifugal pump. What about the effects of particle concentration, and particle density on the erosion distribution/rate?

- Figure 13. Distribution of erosion rate on impeller with different particle diameters – the results obtained for the different particle diameters look quite similar. Maybe you can look also at the effect of particle property on performance, by adding some figures, for example, the graphs of Head and Efficiency vs diameter to show the effect of changing the particle diameter.

- Figure 15. The erosion amount change under different particle diameters – for the legend, is the red dots correspond to the Blade outlet? Please check.

Author Response

Dear Editors and Reviewers:

Thank you for your letter and for the reviewers’ comments concerning our manuscript entitled “Numerical prediction of erosion based on the solid-liquid two-phase flow in a double-suction centrifugal pump” (ID: jmse-1265719). Those comments are all valuable and very helpful for revising and improving our paper, as well as the important guiding significance to our researches. We have studied comments carefully and have made correction which we hope meet with approval. Revised portion are marked in the paper. The main corrections in the paper and the responds to the reviewer’s comments are as flowing:

  1. Response to comments 1:

The quality check process has been added in the paper.

  1. Response to comments 2:

The reference section has been updated.

  1. Response to comments 3:

The mathematical model for the solid-liquid two-phase flow has been added.

  1. Response to comments 4:

The reference for the experimental data of energy characteristics are provided by Andritz company and Tsinghua University has been added in the attached file. Due to the requirements of Party A, relevant information needs to be kept confidential and cannot be displayed in the paper.

  1. Response to comments 5:

The reviewer’s comments are very in-depth, I also studied the effects of particle concentration, and particle density on the erosion distribution/rate, which will be published at another paper. The particle concentration and particle density have large influence on the erosion distribution/rate.

  1. Response to comments 6:

The influence of particle diameters on pump performance have been added in the paper.

  1. Response to comments 7:

I have corrected the figure 15.

These are revised replies based on the comments. I hope to get more suggestions

Thank you and best regards.

Yours sincerely,

Round 2

Reviewer 2 Report

The authors have satisfactory addressed my comments, and I therefore recommend the publication of this article in JMSE.